# A framework for identifying the recent origins of mobile antibiotic resistance genes

Stefan Ebmeyer[1,2], Erik Kristiansson[1,3] & D. G. Joakim Larsson [1,2✉]

Since the introduction of antibiotics as therapeutic agents, many bacterial pathogens have developed resistance to antibiotics. Mobile resistance genes, acquired through horizontal gene transfer, play an important role in this process. Understanding from which bacterial taxa these genes were mobilized, and whether their origin taxa share common traits, is critical for predicting which environments and conditions contribute to the emergence of novel resistance genes. This knowledge may prove valuable for limiting or delaying future transfer of novel resistance genes into pathogens. The literature on the origins of mobile resistance genes is scattered and based on evidence of variable quality. Here, we summarize, amend and scrutinize the evidence for 37 proposed origins of mobile resistance genes. Using state-of-the-art genomic analyses, we supplement and evaluate the evidence based on well-defined criteria. Nineteen percent of reported origins did not fulfill the criteria to confidently assign the respective origin. Of the curated origin taxa, >90% have been associated with infection in humans or domestic animals, some taxa being the origin of several different resistance genes. The clinical emergence of these resistance genes appears to be a consequence of antibiotic selection pressure on taxa that are permanently or transiently associated with the human/domestic animal microbiome.

[1] Center for Antibiotic Resistance Research, University of Gothenburg, Gothenburg, Sweden. [2] Department of Infectious Diseases, Institute of Biomedicine, Sahlgrenska Academy, University of Gothenburg, Gothenburg, Sweden. [3] Department of Mathematical Sciences, Chalmers University of Technology and University of Gothenburg, Gothenburg, Sweden. ✉email: joakim.larsson@fysiologi.gu.se

Antibiotic resistance has become a major threat to the fundaments of modern medicine during the last decades. Resistance genes have emerged against almost all classes of antibiotics, even against those considered last resort. Mobile antibiotic resistance genes (ARGs) are associated with a variety of mobile genetic elements (MGEs) that enable these genes to spread to new hosts, even across taxonomic boundaries. Insertion sequences (ISs)[1,2] and IS common region (ISCR) elements[3–5] have been shown to provide both mobility and strong promotors for the expression of ARGs. In several cases, these are key elements for the mobilization of ARGs from a bacterial chromosome to transferable MGE, such as plasmids or conjugative transposons[6–8]. Although novel ARGs frequently are reported, their origins, meaning the bacterial taxa from which these genes were mobilized, facilitating their transfer to clinically relevant mobile vectors, are unknown for the largest part. Understanding where ARGs come from and which environments, conditions, and/or human practices favor their emergence is necessary to effectively mitigate the emergence of still unknown ARGs in the clinics. Such knowledge will, however, likely be of limited use for managing the ARGs that are already found in the clinics, as it has been shown that once emerged, ARGs are maintained in parts of a population for a long time even in absence of selection pressure, making their emergence practically irreversible[9,10].

Predicting the conditions that are likely to favor the emergence of new ARGs based on the known origins of single or a few ARGs is difficult. However, identifying the origins of a greater number of ARGs may enable us to recognize underlying patterns, such as shared characteristics of origin species, how ARGs are mobilized from their most recent non-mobile origin, the environments they thrive in or their connections to human- or animal associated bacteria. A strong hypothesis is that selection pressure from antibiotic use in humans and domestic animals has played a critical role in their emergence[11,12]. As the environmental reservoir of ARGs is much greater than what is identified in the human and domestic animal microbiota, it is possible that other environments, as well as anthropogenic selection pressures in those environments, play a critical role in this development too[13,14]. Although the origins of some mobile ARGs have been reported and extensive reviews exist on the origin and dissemination of specific groups of clinically problematic ARGs such

as mobile *AmpC* enzymes[15,16] or CTX-M family β-lactamases[17], there is no summary or analysis of all proposed ARG origins to date. Furthermore, the types and quality of evidence presented to identify those origins vary substantially, from mostly molecular methods to (more recently) purely sequencing-based approaches. This creates the need to carefully evaluate each reported ARG origin based on a fixed set of criteria, to ensure the integrity of our knowledge about ARG origins and guide future efforts to identify these.

In this study, we establish a set of comparative criteria that can be used to identify the origins of mobile ARGs on at least genus level with high confidence, based on patterns recognized from thorough literature research. We evaluate the evidence on each previously reported origin based on these criteria and supplement missing data (if available) through amendment with publicly available genome data and state-of-the-art comparative genomics analysis. Finally, we analyze the existing data and discuss what we can conclude from the curated list of origins with regards to overarching patterns on origin species traits and taxonomy, as well as mode of ARG mobilization from their origin and potential circumstances favoring their emergence in human pathogens.

Scrutinizing previous reports, we find that only 81% of suggested origins are supported by the data. We show that all confirmed origin taxa are Proteobacteria, several species being the origin of several ARGs and almost all associated with infection in humans or domestic animals. These results highlight the potential importance of the human/animal body as mobilization hotspots for ARGs. Still, the lack of known origins for the great majority of ARGs points towards unsequenced environmental bacteria as likely sources.

## Results

Our literature search yielded 3342 articles. After screening of all hits (including the references of articles deemed relevant), 43 articles providing evidence for genus or species origins of an ARG were retained for further analysis, which is illustrated in Fig. 1.

**Definition of origin**. For mobile ARGs found in pathogens, we use the term recent origin, describing the bacterium belonging to the evolutionary most recent taxon where the ARG is widespread,

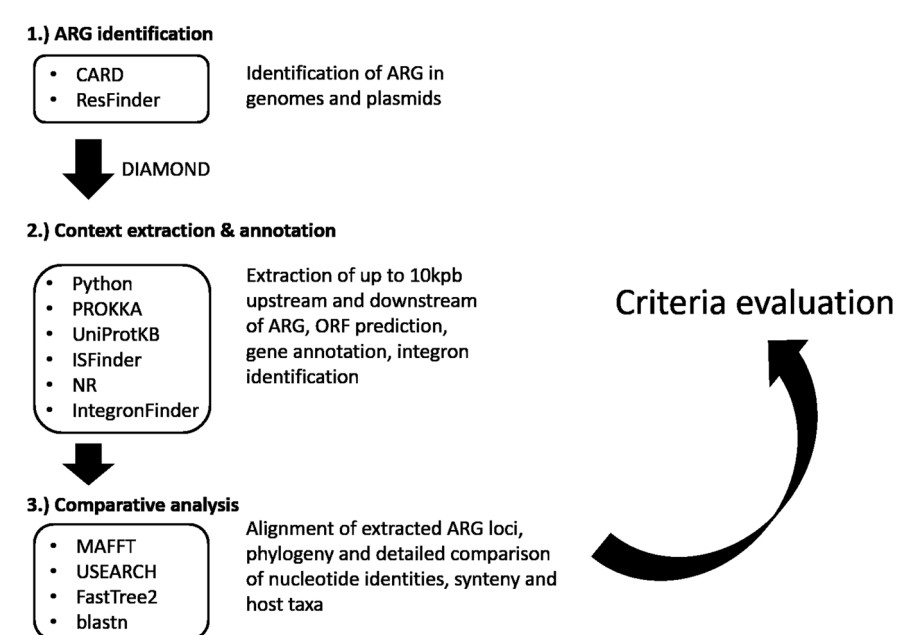

**1.) ARG identification**

- CARD
- ResFinder

Identification of ARG in genomes and plasmids

DIAMOND

**2.) Context extraction & annotation**

- Python
- PROKKA
- UniProtKB
- ISFinder
- NR
- IntegronFinder

Extraction of up to 10kpb upstream and downstream of ARG, ORF prediction, gene annotation, integron identification

**3.) Comparative analysis**

- MAFFT
- USEARCH
- FastTree2
- blastn

Alignment of extracted ARG loci, phylogeny and detailed comparison of nucleotide identities, synteny and host taxa

Criteria evaluation

**Fig. 1 Comparative genomics workflow.** Workflow and tools/databases used for amendment and scrutinization of proposed origins of ARGs.

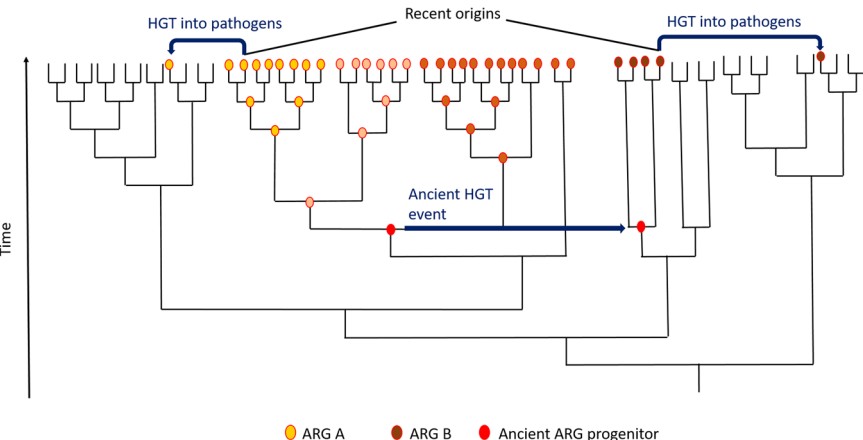

**Fig. 2 The principle of recent origins.** Schematic phylogeny illustrating the principle of recent origins of ARGs. Blue arrows represent horizontal gene transfer events (HGT); red circle on node represents an ARG progenitor. Changing color of circle represents sequence evolution over time. Two possible scenarios are shown: ARG A evolves in the same taxonomic clade as the ARG progenitor prior to being transferred to a pathogen. In case of ARG B, the ARG progenitor is acquired through an ancient HGT event before it is, more recently, transferred to pathogens, and is thus not present in the sister clades of the recent origin of ARG B. In both cases, the recent origin is the taxon from which the gene is mobilized into clinically relevant contexts.

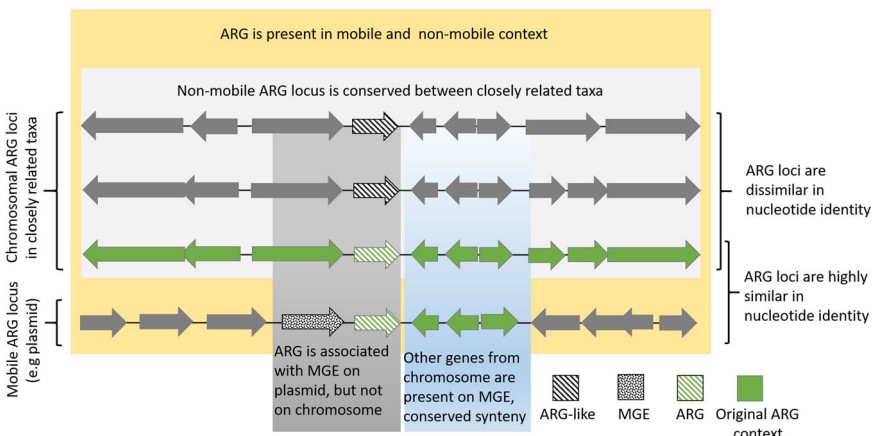

**Fig. 3 Assigning the recent origin of an ARG.** Graphical representation of strong evidence for assigning the recent origin of an ARG using the evaluation criteria described above.

but commonly not associated with (any of) the MGE(s) (e.g., IS, integron, transposon, plasmid, or as part of an integron) that played a role in the ARGs transition to its clinically relevant contexts. Thus, recent origin does not encompass the evolutionary processes that shaped the genes functionality, whether ancient[18] or recent[17], nor ancient transfers across taxa involving different mechanisms. This distinction is important, because there is evidence that the recent origins of some genes may not necessarily be the taxonomic context in which the respective gene originally evolved. For guiding mitigations, however, it is most important to understand from which bacteria ARGs have been mobilized prior to being transferred to pathogens, processes where anthropogenic activities could have played a role. The mobile *qnrB* gene, e.g., shown to have been mobilized to clinically important plasmids from the chromosome of *Citrobacter* spp., is found in several but not all species of the genus. Yet, their genetic environment is very similar in all species harboring the gene on their chromosome[19]. Similar findings have been reported for PER-type[8] and *qnrE*[20] genes. This suggests that certain lineages of these genera may have acquired the respective genes at some point during their more ancient evolution, before they were mobilized to the MGEs that we observe in clinical isolates today (see Fig. 2).

**Formulation of evaluation criteria**. By scrutinizing previously proposed origins and thorough reflection on the strength of different lines of evidence, we formulated the following criteria to identify the recent origin of an ARG (Fig. 3):

- The mobile ARG is associated with a recognizable genetic element that may provide mobility, such as a plasmid, transposon, IS, or ISCR element.
- In the proposed origin taxon, the ARG is found in a genetic context that does not appear transferable by common mechanisms, such as association with plasmids, transposons, etc.
- There is conserved synteny (the genes in the intermediate vicinity of the ARG are arranged in the same/similar order) between the origin chromosome and the mobile ARG locus (due to co-mobilized genes). This criterion excludes genes potentially providing mobility (such as IS) and is rarely applicable for gene cassettes, as usually no surrounding genes are co-mobilized.
- There is high nucleotide identity between the mobile ARG, including the complete mobilized sequence, and the native ARG locus (the higher identity, the stronger the evidence). For confident recent origin genus or species

assignment, nucleotide identities we used ≥90% and ≥95%, respectively.

- The synteny of the ARG locus is to some extent conserved in several species of the origin genus, but the ARG loci are not identical with respect to nucleotide identity.

It is noteworthy that functionality—the ability of a gene to provide resistance—is not part of the criteria, as it is a blunt tool compared to sequence-based analysis. Indeed, several ARGs have been shown not to confer resistance in their original host under the investigated conditions[21,22].

**The majority of confirmed origins are pathogenic non-producers.** We have scrutinized the evidence on the origin of 37 groups of ARGs, which, to the best of our knowledge, summarizes the current literature on specific origins of ARGs. For each ARG group, a detailed description and discussion on the evidence or lack thereof for its proposed origin is provided in supplementary note 1. Based on the criteria set up here, 30 (81%) of the proposed origins could be confirmed for the respective ARG group, though we had to supplement data from our own analysis for most to fulfill all criteria. In seven cases (19%), the criteria were not sufficiently fulfilled to confidently assign an origin (MCR-4, MCR-9, MCR-8, MCR-3, KPC, QnrS, and TetX detailed information in Table 1 and Supplementary Note 1 under the respective paragraph). Further data would be needed to confirm or correct the reported origins of those resistance genes. An increase in the taxonomic resolution of one of the reports through our analysis identified the recent origin of *qnrE* as *Enterobacter mori*, where previously only an origin in *Enterobacter* spp. was reported (see Supplementary Note 1—*qnrE* from *Enterobacter* spp.). The origins of 27 groups of ARGs could be confirmed on species level, resulting in 22 species groups, all of those being Proteobacteria. None of the here reported recent origins are known antibiotic producers (although this was expected for, e.g., the origins of *qnr* genes, as fluoroquinolones are synthetic antibiotics not produced in nature). The respective mobilized genes encode resistance to five different classes of antibiotics: aminoglycosides (6.8%), β-lactams (62%), fluoroquinolones (13.7%), fosfomycin (13.7%), and colistin (3.4%) (see Table 1).

Interestingly, nearly all origin species have previously been associated with humans and/or domestic animals. Out of 22 origins identified at species level, 21 (95%) have been isolated from humans or domestic animals, the exception being *Rheinheimera pacifica*, the likely origin species of LMB-1. For comparison, out of 100 randomly selected proteobacterial species with available genomes, 28 species have been isolated from humans or domestic animals, whereas 72 were not (Fisher's exact test odds ratio: 54.0, p = 3.08e − 9).

More strikingly, all of these origin taxa (except *R. pacifica*) have been isolated from infection sites. For comparison, out of the 100 randomly selected proteobacterial species, 24 species have been isolated from infection sites, whereas 76 have not (Fig. 4). Based on this subsample of proteobacterial species, Fisher's exact test showed that the odds of being an origin were higher for species that have been reported in infection compared to species that never have been reported in infection (Fisher's exact test odds ratio: 66.5, p = 3.13e − 10).

**More ARGs are associated with transposable non-integron elements than with integrons.** None of the genes for which origins have been confirmed is a gene cassette and the evidence suggests in most cases that the genes have been moved from their original context by IS or ISCR elements. Our analysis of associations of ARGs with different modes of mobility (gene cassette vs. transposable unit (excluding mobile integrons)-associated) resulted in 144 different genes belonging to 63 protein families being classified

as gene cassettes, whereas 512 genes from 264 families were classified as associated with transposons. Although we can identify gene cassettes from clinically relevant integron classes with relatively high confidence, the estimate on transposon-borne ARGs is much more speculative, as it is difficult to say whether the respective IS/ISCR/transposon truly provides mobility to the respective ARG, or whether it is just present in close vicinity due to the highly mosaic nature of many ARG-bearing MGEs.

## Discussion

The criteria proposed in this study for identifying the origins of a mobile ARG are a reflection of what we found in the literature—many studies describing the recent origins of mobile ARGs paint a consistent picture: the mobile ARG is usually associated with a genetic element providing mobility, whereas such an element is not present at the presumed non-mobile ARG locus. In nearly all cases, one or several genes from the non-mobile genetic environment of the ARG have been co-mobilized to the mobile ARG locus and the nucleotide identity between the non-mobile and the mobile ARG locus is high, including non-coding regions. The identity cutoffs for genus/species origins specified in criterion 4 are based on the observations made in this study (see Table 1). The presence of the gene locus (although somewhat divergent) in closely related species helps to establish whether the locus has been associated with the respective species chromosome for some time. The majority of the criteria apply in all cases where there is strong evidence for the origins of an ARG—which is why we propose it should be a standard procedure to check for these criteria when investigating a species as origin of an ARG. It should however be noted that in some cases not all criteria would have to be fulfilled to assign an origin—a gene cassette for example will rarely be associated with co-mobilized genes.

All of the to date, with high confidence, identified origin species are Gram-negative Proteobacteria and have been, with the exception of one species, reported in infections of humans and/or domestic animals. Many of these species are well-studied human and animal pathogens or commensals, such as *Citrobacter freundii*, *Klebsiella pneumoniae*, or *Acinetobacter baumanii*. Even more intriguing is that many species/genera are the origin of not a single, but often several different ARG families. *K. pneumoniae* is, e.g., not only the recent origin of SHV-type β-lactamases[23] but also of the mobile *oqxAB* efflux pump[24] and the *fosA5/6* fosfomycin resistance genes. *Kluyvera georgiana* is the recent origin of both the CTX-M-8/9 cluster of class A β-lactamases[25] and the fosfomycin resistance genes *fosA3/4*[26], whereas the *C. freundii* complex is the recent origin of the *qnrB*[19] and CMY-2[27] family genes. Our analysis suggests that a proteobacterial species is much more likely to be the origin of an ARG if it has been isolated from an infection site in humans or domesticated animals. The overrepresentation analysis shows that this finding is not simply a product of database bias towards pathogenic species, as the great majority of randomly sampled proteobacterial species in the NCBI Assembly database have not been reported in infection. Although this does not at all exclude the possibility of other environmental bacteria (meaning species more often associated with external environments) serving as origin species, this data strongly suggests that the potential of a bacterial species to be a recent origin of an ARG is linked to its ability to colonize humans or domesticated animals, at least for proteobacterial species. In these environments, those species may experience severe antibiotic selection pressure and at the same time encounter MGEs that have mobilized ARGs in the past. As previously mentioned, there is also convincing evidence that in many cases different variants of the same ARG family have emerged independently from one another, as opposed to being the result of post-mobilization evolution. This includes the CMY-1/MOX-1 AmpC enzymes from

**Table 1 Evidence for proposed origins of specific ARGs.**

| Resistance determinant | Origin taxon | Antibiotic class | Co-mobilized genes | Nucleotide identity MGE/origin | IS/ISCR on MGE | MGE in proposed origin | ARG loci in origin-related species | Conclusive evidence | Reference |
|---|---|---|---|---|---|---|---|---|---|
| APH(3')-IV | Acinetobacter guillouiae | Aminoglycosides | 1 | 95-98%[a] | ISAba125, ISAba14 | Absent | Yes | Yes | Yoon et al.[21] |
| AAC(6')-Ih | Acinetobacter gyllenbergii | Aminoglycosides | >1 | 98-≥99%[a] | ISAba23, ISAcsp5 | Absent | Yes | Yes | Yoon et al.[60] |
| FOX | Aeromonas caviae | β-Lactams | 1[a] | ≤78%[a] | IS26, ISAs2, Tn3-like[a] | ISApu2Δ | (Yes)[b] | No | Fosse et al.[61] |
| FOX | Aeromonas allosaccharophila | β-Lactams | >1 | 95-98% | IS26, ISAs2, Tn3-like[a] | Absent | Yes | Yes | Ebmeyer et al.[62] |
| CMY-1/MOX-1 | Aeromonas sanarelli | β-Lactams | >1 | 97-98% | ISCR1 | Absent | Yes | Yes | Ebmeyer et al.[28] |
| MOX-2 | A. caviae | β-Lactams | 1 | 91-99% | ISKpn9 | Absent | Yes | Yes | Ebmeyer et al.[28] |
| MOX-9 | Aeromonas media | β-Lactams | 138 bp upstream | 98-99% | ISKpn9 | Absent | Yes | Yes | Ebmeyer et al.[28] |
| CMY-2-like | Citrobacter freundii | β-Lactams | >1 | ≥98% | ISEcp1 | Absent | Yes | Yes | Wu et al.[27] |
| DHA | Morganella morganii | β-Lactams | 1 | ≥97% | Unknown/none detected[a] | Absent | Yes | Yes | Barnaud et al.[63] |
| ACT-1 | Enterobacter asburiae | β-Lactams | 1 | 95-96% | Unknown/none detected[a] | Absent | Yes | Yes | Rottman et al.[64], Reisbig et al.[30] |
| MIR-1 | Enterobacter cloacae | β-Lactams | 1 | >99%[a] | ISPps1[a] | Absent | Yes | Yes | Conceicao et al.[31], Jacoby et al.[15] |
| ACC | Hafnia alvei/paralvei | β-Lactams | 1 | 82->99%[a] | ISEcp1 | Absent | Yes | Yes | Nadjar et al.[65] |
| SHV | Klebsiella pneumoniae | β-Lactams | >1 | ≥99%[a] | IS26, IS102 | Absent | Yes | Yes | Ford et al.[23] |
| OXA-23 | Acinetobacter radioresistens | β-Lactams | 1 | 98->99%[a] | ISAba1, ISAba4 | Absent | Yes | Yes | Poirel et al.[22] |
| OXA-48/181 | Shewanella xiamenensis | β-Lactams | 1 | 100% | ISEcp1 | Absent | Yes | Yes | Potron et al.[66] |
| OXA-51-like | Acinetobacter baumannii | β-Lactams | >1 | >99%[a] | ISAba1 | ISAba1/absent | Yes | Yes | Chen et al.[33] |
| PER | Pararheinheimera spp. | β-Lactams | >1 | 78-96% | ISPa12,ISPa13, ISCR1 | Absent | Yes | Yes | Ebmeyer et al.[8] |
| CTX-M-8/9/25 | Kluyvera georgina | β-Lactams | 1 | 99% | IS10, ISEcp1, ISCR1 | Absent | Yes | Yes | Poirel et al.[25], Rodriguez et al.[67] |
| CTX-M-1,2,3,4,5,6,7 | Kluyvera ascorbata | β-Lactams | 1 | 100% | ISEcp1, ISCR1 | Absent | Yes | Yes | Humeniuk et al.[68], Rodriguez et al.[69] |
| LMB-1 | Rheinheimera pacifica | β-Lactams | 1 | 99%[a] | IS6, IS91 | Absent | (Yes) | (Yes)[b] | Lange et al.[70] |
| KPC | Chromobacterium spp. | β-Lactams | None identified | ≤76% | Tn3-like (Tn4401) | Absent | (Yes)[b] | No | Gudeta et al.[71] |
| GPC-1 | Shinella spp. | β-Lactams | None identified | 89% | IS91, tnpA | Absent | Yes | (Yes)[b] | Kieffer et al.[42] |
| BKC-1 | Shinella spp. | β-Lactams | None identified | 87% | ISKpn23 | Absent | Yes | (Yes)[b] | Kieffer et al.[42] |
| MCR-2 | Moraxella pluranimalium | Colistin | 1 | 96%[a] | IS1595 | Absent | Yes | Yes | Poirel et al.[72], Kieffer et al.[73] |
| MCR-4 | Shewanella frigidimarina | Colistin | None identified | 100% | IS5 | Tn5044 | Not identified | No | Zhang et al.[74] |
| MCR-3 | Aeromonas spp.[b] | Colistin | ~1 | 85-95% | ISKpn3, ISAs17, TnAs2 | Different IS at conserved locus | Yes | No | Yin et al.[75], Shen et al.[76], Khedher et al.[77] |
| MCR-8 | Stenotrophomonas | Colistin | None identified | ≤63% | IS903B, ISEcl1 | Absent | /[b] | No | Khedher et al.[77] |
| MCR-9 | Buttiauxella spp. | Colistin | 1 | 82% | IS26 | Absent | Yes | No | Kieffer et al.[78] |
| QnrB | C. freundii | Fluoroquinolones | >1 | ≥97% | ISCR1, ISEcp1, IS3000, IS6100, IS26 | Absent | Yes | Yes | Jacoby et al.[19], Ribeiro et al.[79] |
| QnrA | Shewanella algae | Fluoroquinolones | >1 | ≥97%[a] | ISCR1 | Absent | Yes | Yes | Poirel et al.[47] |
| QnrE | Enterobacter spp./E. mori | Fluoroquinolones | >1 | 83-≥99%[a] | ISEcp1 | Absent | Yes | Yes | Albornoz et al.[20], This article |
| QnrS | Vibrio splendidus | Fluoroquinolones | None identified | ≤79%[a] | IS2[a] | Unknown | Yes | No | Cattoir et al.[80] |
| OqxAB | K. pneumoniae | Fluoroquinolones | 1 | 97->99%[a] | IS26 | Absent | Yes | Yes | Kim et al.[24] |
| FosA1 | E. cloacae/Enterobacter spp.[b] | Fosfomycin | 1 | 88-99%[a] | Tn2921, IS4 | Absent | Yes | Yes | Ito et al.[81] |
| FosA5/6 | K. pneumoniae | Fosfomycin | >1 | ≥99% | IS10, IS1, IS26 | Absent | Yes | Yes | Ma et al.[82] |
| FosA3/4 | Kluyvera georgiana | Fosfomycin | >1 | ≥99% | IS26, ISEcp1 | Absent | Yes | Yes | Rodriguez et al.[26], Ito et al.[48] |
| FosA8 | Leclercia adecarboxylata | Fosfomycin | >1 | ≥99% | Unknown/none detected[a] | Absent | Yes | Yes | Poirel et al.[83] |
| TetX | Sphingobacterium spp. | Tetracyclin | None identified | ≥99% | Tn6031 | Different mob genes, integrases, transposases | Not identified | No | Ghosh et al.[84] |

[a]Data added through this study.
[b]See information on respective resistance determinant in Supplementary Note 1 for details.

*Aeromonas* spp.[28], the PER ESBLs from *Pararheinheimera* spp.[8] (Fig. 5, *bla*PER-6 locus as reported by Nordmann et al.[29]), the SHV and FosA5/6 genes from *K. pneumoniae*[23], the ACT/MIR AmpC cephalosporinases from *Enterobacter cloacae/asburiae*[30,31], *qnrB* from *Citrobacter* spp., different variants of OXA-51-like genes from *Acinetobacter baumannii*[32,33], CTX-M enzymes from *Kluyvera* spp.[17], and more with variable levels of evidence. This further supports the hypothesis that these recent origin species can thrive in habitats repeatedly exposed to strong antibiotic selection pressure by different classes of antibiotics, such as the human/domesticated animal microbiome. A limitation for making broader generalizations is that most of the origins described here relate to genes that confer resistance to β-lactams. Thus, the patterns described here may or may not reflect genes providing resistance to other antibiotic classes, where no origin is known, to the best of our knowledge (e.g., tetracyclines or macrolides). Thus, identifying their origin using this set of criteria may not be possible. Importantly, the NCBI Assembly database is biased towards proteobacteria—it is entirely possible that in the future, as the diversity of available genomes increases, non-proteobacterial, non-infection-associated

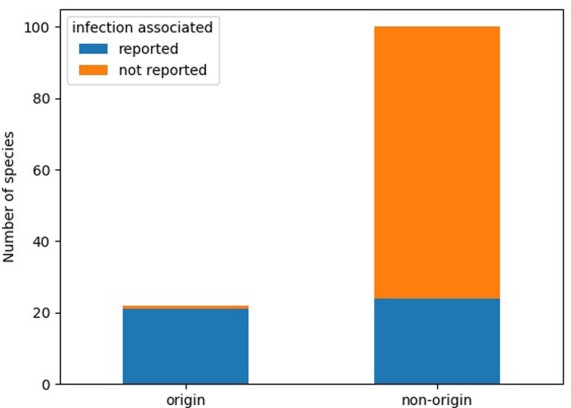

**Fig. 4 Association with infection for randomly selected Proteobacteria and origin species.** Grouped barchart showing number of species reported in infection for origin and non-origin species ($n = 122$). Non-origin species were randomly selected from all proteobacterial species in the NCBI Assembly database. Fisher's exact test odds ratio: 66.5, $p = 3.13e - 10$.

recent origins of ARGs are identified. Indeed, there is some evidence for the dissemination of clinically relevant ARGs from Actinobacteria to Proteobacteria[34]. However, to date reported sequence similarities between non-mobile actinobacterial genes and proteobacterial ARGs are too low (<70% amino acid identity) to draw any conclusion about Actinobacteria as recent origins of these genes—possibly due to lack of the relevant actinobacterial genomes.

There are several factors that have to come together for the successful emergence of a mobilizable ARG: in addition to sufficiently strong antibiotic selection pressure, MGEs with the potential to mobilize genes, such as IS/ISCR, also have to be present, as well as compatible recipients for the newly mobilized ARG. The simplest setting where all of these factors come together is the human/animal body during antibiotic treatment. The human/animal microbiome likely already contains many IS/ISCR elements that have been suspected to be involved in ARG mobilization[35,36] and the constituents of these microbiomes engage intensively in horizontal gene transfer[37,38], especially under stress. Some of the species reported to be recent origins of ARGs are members of the human microbiome, such as *Enterobacter*, *Klebsiella*, *Citrobacter*, or *Acinetobacter*[39]. Others are opportunistic pathogens that are known to at least pass the human/animal body, sometimes causing infection and thus directly triggering antibiotic use, such as *Aeromonas* spp. Other genera, such as *Rheinheimera* and *Pararheineheimera* have not yet been reported to be part of the human/domestic animal microbiome or to be involved in disease. However, as both genera can be found in water (both fresh and saline), it is still possible that these bacteria may pass the human/animal body through ingestion of contaminated water. Although it is certainly possible for a novel ARG to be mobilized in the external environment and be transferred to human pathogens later, this scenario involves more steps and may therefore be less likely, especially if strong antibiotic selection pressure is required for the emergence process. Capture of ARGs from bacteria that are temporarily passing through the body might be possible, given that we know the recent origins of only a small percentage of all known ARGs. We mostly know these origins, because many of the species have at least some clinical relevance and our efforts to characterize (and in the last decade, sequence) have been biased heavily towards bacteria frequently causing disease. Possibly, the great majority of ARG origins is unknown, because they are not permanently associated with the human/animal body, rarely or never cause disease and have

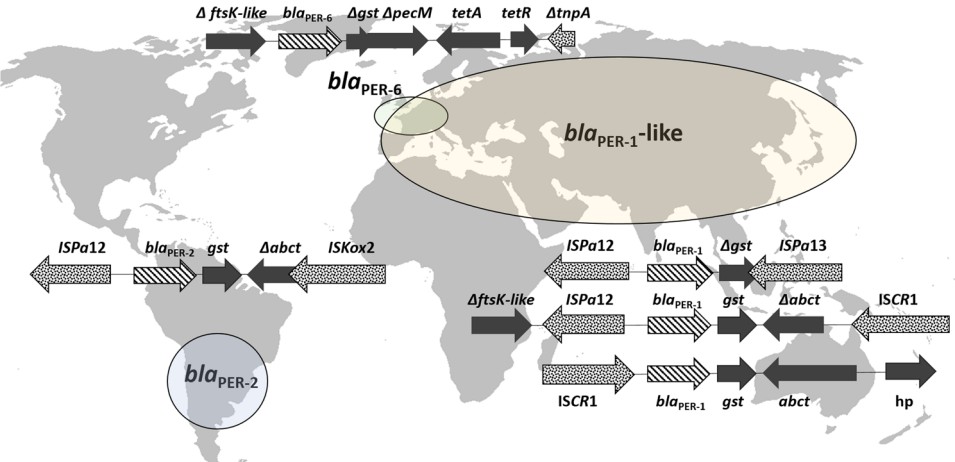

**Fig. 5 Multiple mobilization events of an ARG may result in association with different mobile contexts.** Multiple contexts for ARGs of the same family as a consequence of multiple independent mobilization events on the example of *bla*PER ESBLs, mobilized from *Pararheinheimera* spp. Elliptic shapes (grossly) represent the areas from which different *bla*PER variants have been reported. Although *bla*PER-2 and *bla*PER-1 are both associated with IS*Pa12*, the *bla*PER loci (including *bla*PER and co-mobilized genes) on the depicted MGEs are about 20% dissimilar in nucleotide identities. *bla*PER-6 is 13% dissimilar to *bla*PER-2 and about 20% dissimilar to *bla*PER-1. Transposable elements are presented with dotted pattern and *bla*PER genes with striped pattern.

therefore not been investigated extensively (or are completely unknown to science yet). The ResFinder database contains 3095 unique ARGs up to date (September 2019), corresponding to 696 gene clusters when clustered at 90% nucleotide similarity. Based on this and the origins described here, we currently know the origin of only about 4% of described acquired resistance genes. For some classes of ARGs, such as tetracyclines or macrolides, no species origin is known at all, despite tetracyclines being one of the most extensively used antibiotics in, e.g., agriculture[40]. This suggests that many ARG origins thrive in other environments, such as water, soil, or associated with plants, underlining that the natural environment may act as a source of novel resistance determinants[41,42] (as supported by, e.g., functional metagenomics studies identifying many novel ARGs from soil, see Forsberg et al.[43]). Such environmental species may pass through the body from time to time by exposure to contaminated matter, where their genome, especially under antibiotic selection pressure, may be infiltrated by MGEs from the hosts microbiome, such as plasmids, IS/ISCRs, transposons, or integrons. However, even for described origin genera/species such as *Kluyvera* spp, *Pararheinheimera* spp, and *Aeromonas* spp, knowledge on where exactly these bacteria thrive is lacking and further research is needed on the topic. Another explanation for the lack of origin species for some classes of ARGs may be that they have been mobilized a long time ago, so their origin has evolved and cannot be identified anymore. Verifying this hypothesis, however, requires more genomic data than available to date, especially from bacteria not associated with the human or domestic animal microbiome.

The accumulation of ARGs in human-associated bacteria may also have implications for the emergence of novel ARGs in human pathogens. As many of the ARGs are associated with IS/ISCR elements, the number of these MGEs at selection sites, e.g., the human/animal body, also increases. This may result in an increased potential of the respective microbiome to capture rare genes (e.g., from bacteria that are only temporarily associated with those microbiomes), which can help to respond to local selection pressures, using highly effective capturing machineries, such as ISCR1, IS26, or ISEcp1. This is especially true for IS26, as it has been shown that IS26 and co-mobilized sequences target sites associated with other IS26 for integration[44], or ISCR1, which couples newly mobilized ARGs with class I integrons. Thus, increased acquisition of resistance genes in a microbial community may also increase the potential to capture novel ARGs.

Antibiotic selection pressure is likely a key factor during ARG mobilization in many cases. The high nucleotide identities of ARG loci towards the chromosomal loci of their origins, as summarized in this study, suggest that all here described mobilization events have happened relatively recently, quite plausibly during the antibiotic era and driven by human-imposed antibiotic selection pressure[45,46]. In most cases, up- or downstream sequences from the ARGs original context have been co-mobilized with the ARG and they are highly identical in nucleotide sequence to their chromosomal counterparts, but we observe that these co-mobilized sequences are often truncated and thus likely non-functional. If the mobilization of these ARG loci to MGEs were ancient events, higher sequence dissimilarity to their non-mobile counterparts would be expected, if they were preserved at all. Although it has been hypothesized that some ARGs have been on plasmids for millions of years[46] and evolved there, thus potentially lacking a recent origin, the data presented here strongly support that at least some ARGs have a recent chromosomal origin. However, this does not exclude the possibility that these genes have been plasmid-borne at some previous time point during their evolution (see Fig. 2).

Our association analysis indicates that, of the mobile resistance genes found in all to data available genomic assemblies, about four times as many ARG families (at 80% sequence similarity) are associated with transposases, IS, or ISCR compared to ARG families that are integron-borne gene cassettes. Although the estimate on transposon/IS/ISCR association is somewhat gross, this finding still underlines that transposable elements including ISs may be the main disseminating agents of mobile ARGs between different replicons. The summarized literature on ARG origins (Supplementary Note 1, Supplementary References) provides insights as to why it may be like that and it is also as discussed below.

It has been shown that several ARGs do not confer clinical level resistance in their native context. This renders some of the above described origin species susceptible to the respective antibiotic[21,22,47,48], even though they encode a potential resistance gene on their chromosome. Only after increased expression of the respective gene or a transfer of the potential ARG to a multi copy vector, which effectively increases gene dosage, does the gene induce a clinically relevant resistance phenotype. As described in detail in Supplementary Note 1, most of the ARGs with known origins were likely mobilized from their native context by IS or ISCR elements. The ability of those types of MGEs to transpose not only themselves but also adjacent DNA sequences, upon transposition via different mechanisms, has been demonstrated for different types of IS. Especially ISCR1, ISEcp1, and IS26 are associated with different ARGs and large parts of their native context in clinical isolates[8,28,49]. Although it has not been experimentally validated to date, that ISCR1 moves and mobilizes adjacent genes, its frequent association with different ARGs and parts of their native context provides evidence for this hypothesis. As with ISCR1, many other IS/ISCR are also known to increase the expression of adjacent genes through providing a promoter sequence, the creation of hybrid promoters upon insertion or the disruption of regulatory genes at the target site[4,50]. IS/ISCR induced high expression of a native ARG may provide the host with increased resistance to the respective antibiotic selection pressure[2,3] (e.g., during antibiotic treatment), such that the combination of IS/ISCR and ARG is selected for under these circumstances[51]. The genetic context of mobilized ARGs support this notion, as we observe that in many cases the ARG is located directly adjacent of the IS/ISCR and thus in reach of the respective promoters. Thus, selection leads to a frequency increase of this gene combination in the local origin population, increasing the likelihood that it will be mobilized to a transferrable MGE, such as a conjugative transposon or plasmid.

In this study, we summarize, scrutinize, and supplement the current knowledge on the origins of mobile ARGs. The here defined set of criteria enables the identification of ARG origin species with high confidence. All origins identified up to date are Proteobacteria and the great majority have been associated with infection in humans or domestic animals, hence triggering antibiotic use. Most ARG families for which an origin has been identified consist of several variants. In many cases, these are likely the product of multiple, independent mobilization events, as they are associated with different remnants of their native contexts or different mobilizing agents, such as IS or ISCR elements. The most likely environment in which these genes have emerged is the human/animal body subjected to antibiotic treatment, as it fulfills all criteria needed for ARG emergence, especially the one of strong antibiotic selection pressure during the entire process. That said, it is certainly possible that other ARGs have emerged from other environments, via different routes, as most ARG origins are still unknown. Further research is thus needed to show to what extent different environments and conditions contribute to ARG emergence. This includes increased whole-genome sequencing efforts of many more bacterial taxa. As genes whose native function may not primarily be associated with antibiotic resistance can become ARGs under the right conditions, the bacterial pan-genome very likely contains an

enormous number of ARGs that have not (yet) been observed in the clinics. Exploring the origins of ARGs further can provide us with important insights as to where our mitigation efforts may be most effective, to slow down the emergence of novel ARGs in the clinics.

## Methods

**Literature research**. A thorough literature research was conducted using the PubMed database (using the key words "origin," "antibiotic," and "resistance gene"). Hits were screened (by title and abstract) to identify articles claiming to have identified the origins of ARGs on at least genus level and references of identified articles were used to identify further articles of the same type. The evidence presented for the respective origin of an ARG was summarized (see Supplementary Note 1) and a set of criteria for identifying the origin of an ARG with high confidence was assembled (see Results).

Furthermore, for every ARG that had a reported origin, we described how the putative origin was identified, which MGEs, the respective ARG, has been found to be associated with. For validated origin taxa, we also briefly described known habitats of the respective origin genus/species and whether it has been reported in infection in humans or domestic animals (Supplementary Table 1).

**Evaluating the criteria: comparative genomics analysis**. An in-house pipeline was used to supplement the analysis in the original study with available genomic data, if available. Based on the original data and our comparative genomics analysis, we then evaluate if or to what extent the original report has fulfilled the five criteria for assigning an origin. The pipeline was written in python 3.7 and is described in the following paragraphs.

Available assemblies of bacterial genomes and plasmids were downloaded from the NCBI Assembly database[52] (April 2020). To create a well-annotated ARG reference database containing only acquired ARGs, sequences from CARDs[53] protein homolog model (v3.0.5, downloaded September 2019, recently characterized ARGs LMB-1, FosA8, FosL, and GPC-1 were manually added) was mapped against the Resfinder database[54] (downloaded September 2019) using DIAMOND at a 99% identity threshold and a 90% query coverage cutoff. Using the resulting database, all genomes and plasmids were searched for the ARGs using DIAMOND blastx v0.9.24.125 with a 70% identity cutoff and an 80% query coverage cutoff. This relaxed cutoff was used to also identify potentially related genes in other taxa. For those genes for which an origin was reported, up to 10 kbp (if available) were extracted upstream and downstream of each thus identified ARG and annotated using Prodigal v2.6.3[55]. To identify potential functions of open reading frames (ORFs) in the identified ARGs genetic environment, identified ORFs were also searched against the Uniprot knowledge base (downloaded January 2019, hypothetical proteins were removed) using DIAMOND with a 60% identity cutoff. To reduce the number of uninformative sequences (with regards to mobility/non-mobility) for further analysis, sequences containing fewer than 6 predicted ORFs were excluded. Genes close to identified ARGs that were annotated as transposases or IS were manually searched against the ISFinder database to confirm their identity towards known IS elements. IntegronFinder v2[56] was used to predict integrons or integron associated elements, such as attC sites, on the artificial ARG contigs. To obtain an overall estimate of the proportion of ARGs that are either gene cassettes (present in integron classes I–III) or mobile through IS/ISCR or other transposons (although the estimate is much more gross for this group than for gene cassettes), genes were placed in one of the groups using the following criteria: ARGs were counted as gene cassettes if they were positioned within the boundaries of a predicted complete intergron, had an attC site encoded within at maxiumum 200 bp (accounting for an attC site length of 55–141 nt and a spacer length of maximum 145 bp between gene cassette and attC site) downstream of the predicted ORF, and were oriented in the opposite direction of the integrase encoding gene. If the ARG was positioned in an incomplete integron (CALIN), an attC site both upstream and downstream (at maximum 200 bp distance within either side of the ORF) of the ARG were required to count it as gene cassette. An ARG was counted as associated with a transposase if the next ORF upstream or downstream of the gene was an IS/ISCR or other transposition element and the gene had not previously been classified as a gene cassette.

To reduce the number of sequences to visualize, duplicates were removed. The remaining extracted sequences were then clustered at a 95% nucleotide identity threshold using USEARCH v8.0.1445[57]. The resulting centroids were aligned using MAFFT v7.310[58] and a phylogenetic tree was created using FastTree v2.1.9[59] (gtr model, default CAT approximation). The Python ete3 package was used to schematically visualize each sequence associated with a node in the previously created phylogeny, allowing for careful visual comparison of the genomic contexts of each gene. Nucleotide or amino acid comparisons between different sequences were conducted using NCBIs online available blast suite.

**Overrepresentation analysis**. To investigate whether bacterial species with the potential to colonize humans or domestic animals were overrepresented among species that are recent origins of ARGs, we used Fisher's exact test. As the set of origin species identified down to species level contained only proteobacterial species ($n = 22$, each species only counted once even if it is the origin of several ARGs), we randomly selected 100 proteobacterial species (excluding origin species) from the NCBI Assembly database. Taxonomy information was downloaded from the NCBI Taxonomy database. A literature search was then performed for all 122 species names against PubMed and Google Scholar, once on its own and again in combination with the term "infect." Hits were screened for information on whether there were reports of isolation of the species from infection sites in humans or domesticated animals (see Supplementary Table 1). A contingency table summarizing the counts was created and analyzed using Fisher's exact test from Pythons scipy library.

**Reporting summary**. Further information on research design is available in the Nature Research Reporting Summary linked to this article.

## Data availability
All genomes used in this study are publicly available at https://www.ncbi.nlm.nih.gov/assembly. The synteny figures used as a first visualization of ARG loci in different genomes are available at https://figshare.com/articles/figure/ARG_syntenies/13084553.

## Code availability
The code used in this manuscript is available at https://github.com/EbmeyerSt/ARG_loci_comparative_pipeline.

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

## Author contributions

S.E. was responsible for conception and design of the work, data collection, data analysis, and interpretation and drafting the article. E.K. was responsible for conception and design of the work and critical revision of the article. D.G.J.L. was responsible for conception and design of the work, critical revision of the article, and main supervision.

## Funding

## Competing interests

The authors declare no competing interests
