## [Peer Review File · Communications Biology]

Reviewers' comments:

Reviewer #1 (Remarks to the Author):

The authors of that study made an in-silico analysis in relation with antibiotic resistance genes, focusing on their origin. The topic is of interest and the authors used adequate tools for it, even though one might admit that their findings overall mirrors what has been previously published by others.

What I mean here is that there is almost no novelty at all, and this report is actually a review of what is already known.

The manuscript is much too long in its present form, with many wordy and tedious statements overall.

The recurrent discussion about clinical/non clinical, pathogen/non pathogen is on my opinion out of scope.

Reviewer #2 (Remarks to the Author):

Summary of the manuscript

In their manuscript entitled "A framework for identifying the recent origins of mobile antibiotic resistance genes" Ebmeyer et al. seek to collate existing knowledge of the recent origins of medically relevant ARGs and use this to develop a pipeline for validating and extending them. The authors establish criteria for defining the recent origin of an ARG, first determining that the ARG is mobilizable by synteny with a mobilizable element, second confirming the origin by establishing lack of a mobilizable element, third and fourth determining origin by high sequence identity of syntenic genes, and fifth validating 3 and 4 across closely related taxa. They apply these criteria to a subset of ARGs and claim that in almost all cases the recent origin is in a human or animal Proteobacterial pathogen, suggesting that human activities are driving ARG spread. The authors end by noting that establishing the origins of ARGs is an important exercise that might allow for mitigation of spread in the future.

Overall impression of the work

The manuscript is well written, interesting, and tackles relatively successfully an important question in antibiotic resistance. I cannot easily think of another source for this type of codified pipeline for approaching and validating the origins of ARGs. Where the manuscript could still benefit is in its explanations for certain decisions and the definitions it uses. For example, why are only some antibiotics and ARGs explored (the β -lactamases, AMEs, MCRs, etc but not CATs or efflux mechanisms?), what is meant by recent ARG origin (evolutionarily recent? Recent since the clinical use of antibiotics?), and others (see specific comments below). While I believe it is important to address these, I overall find the approach and topic of this manuscript compelling.

Specific comments

1. The authors describe their analysis pipeline in good detail and in some cases (e.g. line 101) describe their reasoning for setting numerical cut-offs. However, in other cases (e.g. lines 105, 117) it appears that numerical cut-offs are arbitrary. Could the authors respond why these cut-offs were used and potentially add text to the manuscript that describes how the results of their pipeline change (or don't change) using stricter or more relaxed cut-offs? This would speak to the robustness of the analysis pipeline and the major conclusions of the manuscript.

2. I was surprised and interested to see the conclusion that essentially all ARGs originate from Proteobacterial pathogens when my understanding has been that antimicrobial producing bacteria (largely non-Proteobacterial) are likely the originators of many ARGs (e.g. Benveniste and Davies, 1973; Perry, Waglechner, Wright 2016 and many other Wright lab papers). The authors state in lines 326-331 that they know ARG origins are in Proteobacteria because they are so well studied, but it seems like this could be backwards, that because the databases used have a strong pathogen/Proteobacterial bias this might act as a confound in the analyses.

3. Related to comment #2, the authors could also clarify a bit more what they mean by recent origins of ARGs. Reading the manuscript it seems like the authors change a little in their usage,

going from where an ARG originates in the environment then shifting to where an ARG first becomes mobilized. i.e. an ARG originates from some anonymous soil bacteria but is captured by an *Acinetobacter* sp. which sticks it into a plasmid; which of these two is really the origin depends on a precise definition which is not clearly given. A schematic figure might help?

4. The discussion of mobilizable ARGs seems incomplete without mention or reference to the many findings from functional metagenomic studies that explicitly test mobilization and frequently capture mobilization elements (e.g. Pehrsson et al. 2016 identical TEM bla found across multiple or Forsberg et al. 2014 showing pathogen ARGs to be closely associated with mobilizable elements).

5. Did the authors search for evidence of phage genes as well as evidence for plasmids, transposons, IS, etc? What sort of genomic signal would ARG mobilization by transduction leave and are there examples where this has occurred?

6. Could the authors clarify more how they arrived at this particular set of antibiotic/ARGs to study. It is noted (lines 293 – 296) that tetracycline resistance was omitted due to lack of origin hypotheses, is that the case for tetX homologs as well? I am also curious why chloramphenicol and the widespread cat and phosphotransferase genes are not discussed?

Reviewer #3 (Remarks to the Author):

I read the manuscript from Ebmeyer et al titled "A framework for identifying the recent origins of mobile antibiotic resistance genes" with great interest. The authors have re-evaluated the evidence for claims of the origin of mobile ARGs by appealing to both the literature and examination of available genome synteny and by synthesizing the approaches into a set of criteria. I believe this is an important and timely topic of interest to many readers in antimicrobial resistance and has the potential to be cited by others in this area.

As the manuscript stands, I cannot recommend publication without some additional rigor in the methods and materials section (see specific comments) that would enhance reproducibility and allow interested parties to follow along more closely. In particular, phylogenetic trees were produced, but it is not clear if these are trees of genomic contexts (nucleotide data), or ARG sequences (protein sequences), or both, but there is no mention of these trees in the results. Similarly, virtually none of the genomic synteny results are presented except in summary, making these results difficult to evaluate – can the genomic synteny results be provided as supplementary data? I have no sense for how many loci or different species are represented for each ARG. The discussion firmly situates this work in the larger context of the (recent) origins of ARGs. Given that the ARGs with putatively identifiable origins are restricted to Gram-negative organisms in general, and Proteobacteria specifically, I believe it would be useful for the authors to comment on whether these criteria may be modified to accommodate the 96% of mobile ARGs for which origin cannot currently be assigned.

Specific comments:

Lines 27-33: I would like some additional citations here – I am not familiar with ISCR and it would be helpful to be pointed to some reviews of this topic to orient the reader.

Line 43: Dan Andersson, in particular, has contributed much to this topic in the literature. Of particular relevance for this you may cite Andersson and Hughes *Nat Rev Microbiol* 2010;8(4):260-271.

Lines 73-79: How many articles were identified by this literature search procedure? How many matching articles were retained? The reference list in the supplementary file 1 contains 77 citations. Having performed a similar search in the past, I have noticed that many articles using the phrase 'origin' report no such thing.

Lines 81-86: How often did an article claim to source the origin of an ARG without being associated with a MGE?

Line 92: is the pipeline available for review?

Line 95-96: How were these novel ARGs identified? In the supplementary file?

Line 95: What components of CARD were downloaded? I assume the protein sequences of all

antibiotic resistance determinants? I believe these are also versioned, so which version was used? Similarly, the ResFinder database is also versioned through the bitbucket repository (https://bitbucket.org/genomicepidemiology/resfinder_db.git), which version was used?

Line 97: the 'resulting database' is the intersection of CARD and ResFinder db by alignment, or the union, to include everything in at least one of the databases? I want to understand what the mapping/alignment procedure between the two databases accomplished.

Line 98: to clarify, 80% query coverage cutoff?

Line 99-100: these cutoff criteria are relaxed with respect to the database vs database cutoff criterion, were you only including hits in closely related genera, or more distant taxa? It's very curious to me that the resulting set were restricted entirely to Proteobacteria.

Line 100-102: Genbank assemblies vary greatly with respect to contiguity and completeness – did you have any criteria for excluding highly fragmented assemblies? 10kbp upstream and downstream might not be available in such assemblies, particularly if your goal is to identify ARG associations with IS and Tn sequences which can be highly repetitive and difficult to assemble (and subject to a higher chance of misassembly).

Line 120: FastTree or FastTree2? Which version, and what parameters were used to estimate these trees (substitution model, CAT approximation and categories?) Why produce trees only to never refer to them in the remainder of the manuscript? Were the datasets so large that non-approximate ML were infeasible?

Line 148-149: I appreciate the distinction made between evolutionary time and functional time. I fear some readers might be confused between the origin of a gene, or the origin of a gene as a mobilized ARG which I believe is the intention in this manuscript. There is some debate in the literature between the idea of a 'proto-resistance' gene (Morar and Wright. *Annu Rev Genet* 2010;44:25-51) and a resistance gene being strictly a gene that confers resistance to a clinically relevant antibiotic in the context of a pathogen (Martínez et al, *Nat Rev Microbiol* 2015;13:116-123).

Line 157: I find this idea intriguing, that a potential ARG may be mobilized within a restricted set of taxa

Line 177-179: In your view, are these criteria, particularly #5, likely to hold true outside of Proteobacteria, and what constitutes independence in related taxa? See Pawlowski et al *Nat Comm* 2016;7:13803 for an investigation of the genetic context/synteny of ARGs in *Paenibacillus*.

Line 202: Unsurprising, Proteobacteria are not known as producers of the antibiotics that are or derived from natural products. Fluoroquinolones are synthetic, however, so we would never expect any of these to be producers and this is worth pointing out.

Line 208-209: Given the historically biased nature of the sequence databases towards organisms of medical interest, this isn't surprising.

Line 250-251: I do not know if the authors can say these are correct determinations, only that their criteria support the initial determination of origin.

Line 340: Can you provide an example, citation of this? Most of the examples in Table 1 have origins in a particular species, but several only have origins in a genus. Are the factors preventing inference of origins related to the level of sequence diversity of the ARG, the level of genomic diversity in host species, the degree to which an ARG has proliferated outside of its origin taxa, or something else?

Line 369: all to data -> all to date

Reviewer #1

General comment

1. The authors of that study made an in-silico analysis in relation with antibiotic resistance genes, focusing on their origin. The topic is of interest and the authors used adequate tools for it, even though one might admit that their findings overall mirrors what has been previously published by others.

What I mean here is that there is almost no novelty at all, and this report is actually a review of what is already known.

The manuscript is much too long in its present form, with many wordy and tedious statements overall.

The recurrent discussion about clinical/non clinical, pathogen/non pathogen is on my opinion out of scope.

We respectfully disagree with Reviewer #1 on the novelty of our results. While we indeed summarize the to date scattered literature on the recent origins of ARGs, we use these data to

- 1.) Formulate a set of criteria enabling the confident assignment of resistance gene origins
- 2.) Amend and scrutinize the previously reported results using these criteria, showing that 6 of 29 reports do not have sufficient support for an origin assignment, and improving taxonomic resolution for 1 of the reports
- 3.) Analyze overarching patterns in the scrutinized data - We show that all here verified origin species are not producers of antibiotics, many resistance genes have likely been mobilized several times from their origin, single species are the origin of several different resistance gene families and provide an estimate about association of ARGs with integrons/transposases based on data from all available genome assemblies.
- 4.) Show that species associated with infection in humans and animals are clearly overrepresented as recent origins of antibiotic resistance genes. Therefore, in our opinion the ‘discussion about clinical/non clinical, pathogen/non pathogen’ is a critically important component of the study.

Furthermore, our discussion points out several aspects that have not yet been discussed in the light of the here presented data, such as patterns pointing towards the human/animal microbiome as mobilization hotspots for ARGs from their origin and the importance of identifying the species origins of ARGs correctly in the first place.

Reviewer #2

General comment

Summary of the manuscript

In their manuscript entitled “A framework for identifying the recent origins of mobile antibiotic resistance genes” Ebmeyer et al. seek to collate existing knowledge of the recent origins of medically relevant ARGs and use this to develop a pipeline for validating and extending them. The authors establish criteria for defining the recent origin of an ARG, first determining that the ARG is mobilizable by synteny with a mobilizable element, second confirming the origin by establishing lack of a mobilizable element, third and fourth determining origin by high sequence identity of syntenic genes, and fifth validating 3 and 4 across closely related taxa. They apply these criteria to a subset of ARGs and claim that in almost all cases the recent origin is in a human or animal Proteobacterial pathogen, suggesting that human activities are driving ARG spread. The authors end by noting that establishing the origins of ARGs is an important exercise that might allow for mitigation of spread in the future.

Overall impression of the work

The manuscript is well written, interesting, and tackles relatively successfully an important question in antibiotic resistance. I cannot easily think of another source for this type of codified pipeline for approaching and validating the origins of ARGs. Where the manuscript could still benefit is in its explanations for certain decisions and the definitions it uses. For example, why are only some antibiotics and ARGs explored (the β -lactamases, AMEs, MCRs, etc but not CATs or efflux mechanisms?), what is meant by recent ARG origin (evolutionarily recent? Recent since the clinical use of antibiotics?), and others (see specific comments below). While I believe it is important to address these, I overall find the approach and topic of this manuscript compelling

Specific comments

2. The authors describe their analysis pipeline in good detail and in some cases (e.g. line 101) describe their reasoning for setting numerical cut-offs. However, in other cases (e.g. lines 105, 117) it appears that numerical cut-offs are arbitrary. Could the authors respond why these cut-offs were used and potentially add text to the manuscript that describes how the results of their pipeline change (or don't change) using stricter or more relaxed cut-offs? This would speak to the robustness of the analysis pipeline and the major conclusions of the manuscript.

We agree with the reviewer that further motivations of numeric cutoff values are warranted.

We have added explanations for selection of these cutoffs to the manuscript at lines 103–105 (“To identify potential functions of ORFs in the identified ARGs genetic environment, identified open reading frames (ORFs) were also searched against the Uniprot knowledge base (downloaded January 2019, hypothetical proteins were removed) using DIAMOND [...]”)and lines 116–118 (“[...] (accounting for an attC site length of 55–141nt and a spacer length of maximum 145bp between gene cassette and attC site[...]”). We furthermore clarified that the attC site has to be encoded **within** 200bp downstream of the respective ARG (line 119–121, “[...]at maximum 200 bp distance within either side of the ORF[...]”).

In the referred lines (102–107 in the revised manuscript), a cutoff of 60% is used to compare genes in a ARGs genetic environment to the Uniprot knowledge base. This relaxed cutoff is used in order to get an idea whether a gene is mobile or non-mobile in a respective context, it is most important to find out whether an open reading frame shows homology to genes associated with horizontal gene transfer, such as IS/ISCR, transposases and plasmid mobilization genes, or whether it is more similar to other types of genes. This relaxed cutoff is necessary in order to identify homology over larger evolutionary distances (due to the incompleteness of existing sequence repositories). Increasing this cutoff would lead to a greater number of ORFs annotated as hypothetical proteins, yielding no direct information on their potential function. Lowering this cutoff would increase the number of hits somewhat, but it would increase matches to very distantly related proteins which may have significant differences in their biological function. As described in the materials and method section, to test an origin hypothesis, each ORF in the respective ARGs genetic environment is annotated by manual blast search to confidently identify the genes identity (if possible). So while the first, automated annotation step using the 60% cutoff used in the first part of the pipeline, it is not critical for the robustness of the results, as gene identities have to be confirmed manually.

In line 116–118, the maximum allowed distance between an ORF and a detected attC site was set to 200bp in order for the ORF to be counted as associated with (among other parameters described in the respective paragraph) integron structures of class 1, 2 or 3. As attC sites (itself being 55–141bp long) in these integrons usually are located directly adjacent to the gene cassette, we therefore selected a maximum distance of 200bp to decrease the likelihood of missing the corresponding ORFs while at the same time avoiding as many false positives as possible.

3. *I was surprised and interested to see the conclusion that essentially all ARGs originate from Proteobacterial pathogens when my understanding has been that antimicrobial producing bacteria (largely non-Proteobacterial) are likely the originators of many ARGs (e.g. Benveniste and Davies, 1973; Perry, Waglechner, Wright 2016 and many other Wright lab papers). The authors state in lines 326-331 that they know ARG origins are in Proteobacteria because they are so well studied, but it seems like this could be backwards, that because the databases used have a strong pathogen/Proteobacterial bias this might act as a confound in the analyses.*

Indeed, we cannot exclude that database bias is a confounder and we have expanded on this in the discussion line 318-320 (“Importantly, the Genbank Assembly database is biased towards proteobacteria - it is entirely possible that in the future, as the diversity of available genomes increases, non-proteobacterial, non-infection-associated recent origins of ARGs are identified”). We try to make clear that while we do not know of any non-proteobacterial origin examples, we do not at all exclude the possibility of other bacterial phyla being recent origins of ARGs.

The Genbank assembly database contains about 476600 proteobacterial genomes (about 4911 unique species, excluding incompletely classified ones) and about 20400 *actinobacteria* genomes (about 2529 unique species). Thus, while the database is biased towards proteobacterial species as the reviewer suspected, there is a reasonably large number of actinobacterial species represented as well.

4. *Related to comment #2, the authors could also clarify a bit more what they mean by recent origins of ARGs. Reading the manuscript it seems like the authors change a little in their usage, going from where an ARG originates in the environment then shifting to where an ARG first becomes mobilized. i.e. an ARG originates from some anonymous soil bacteria but is captured by an Acinetobacter sp. which sticks it into a plasmid; which of these two is really the origin depends on a precise definition which is not clearly given. A schematic figure might help?*

We thank the reviewer for an insightful comment. We have had long discussions on how to define “recent origin” and we think we have arrived at a more generally applicable and precise description than the one presented in the original manuscript (lines 146-149 in the original manuscript, lines 158-162 in the revised manuscript). We now define recent origin for a mobile ARG as “.. *the bacterium belonging to the evolutionary most recent taxon where the ARG is widespread, but commonly not associated with (any of) the mobile genetic element(s) (e.g. IS, integron, transposon, plasmid or as part of an integron) that played a role in the ARGs transition to its clinically relevant contexts.*” As suggested, we supplied a schematic figure for clarification purposes (Figure 2 in the revised manuscript).

Fig. 2: Schematic phylogeny illustrating the principle of recent origins of ARGs. Blue arrows represent horizontal gene transfer events (HGT), red circle on node represents an ARG-progenitor. Changing color of circle represents sequence evolution over time. Two possible scenarios are shown: ARG A evolves in the same taxonomic clade as the ARG progenitor prior to being transferred to a pathogen. In case of ARG B, the ARG progenitor is acquired through an ancient HGT event before it is, more recently, transferred to pathogens - and is thus not present in the sister clades of the recent origin of ARG B. In both cases, the recent origin is the taxon from which the gene is mobilized into clinically relevant contexts.

In the provided example, the answer really depends - So if the transfer of the ARG from the soil bacterium to *Acinetobacter* is more ancient and the ARG is fixated on the *Acinetobacter* spp. chromosome over time before it is transferred to a plasmid, the *Acinetobacter* spp. is the recent origin - if the transfer of the ARG from the soil bacterium is more recent and the *Acinetobacter* spp. is just an intermediate 'host' for the already mobilized gene, the soil bacterium would be the recent origin.

5. The discussion of mobilizable ARGs seems incomplete without mention or reference to the many findings from functional metagenomic studies that explicitly test mobilization and frequently capture mobilization elements (e.g. Pehrsson et

al. 2016 identical TEM bla found across multiple or Forsberg et al. 2014 showing pathogen ARGs to be closely associated with mobilizable elements).

We found the articles suggested by the reviewers meaningful and incorporated the findings into our manuscript at line 360–364 (“[...]underlining that the natural environment may constitute a significant source of novel resistance determinants⁴⁷ (as supported by e.g. functional metagenomics studies identifying many novel ARGs from soil , see Forsberg et al 2014⁴⁸). ”).

6. Did the authors search for evidence of phage genes as well as evidence for plasmids, transposons, IS, etc? What sort of genomic signal would ARG mobilization by transduction leave and are there examples where this has occurred?

Our search for mobile genetic elements using the Uniprot knowledge base includes proteins present in phage genomes. However, we did not specifically scan for the presence of phages, as ARGs carried by phages are rare¹ and we do not know of any strong evidence suggesting that phages are important in the mobilization of ARGs.

7. Could the authors clarify more how they arrived at this particular set of antibiotic/ARGs to study. It is noted (lines 293 - 296) that tetracycline resistance was omitted due to lack of origin hypotheses, is that the case for tetX homologs as well? I am also curious why chloramphenicol and the widespread cat and phosphotransferase genes are not discussed?

As stated in line 16–17 of the manuscript, the here discussed set of ARGs is based on ARGs that had proposed origins on at least genus level, identified through a thorough literature research. While, to the best of our knowledge, no such origins have been suggested for cat genes, we realized that we missed to identify an article discussing an origin for tetX genes². We thank the reviewer for pointing this out and we have added tetX to our analysis in this manuscript now (see table 1, supplementary file 1). According to our criteria, there is no reliable evidence for *Sphingobacterium spp* as the proposed origin of tetX genes.

Reviewer #3

General comment

I read the manuscript from Ebmeyer et al titled “A framework for identifying the recent origins of mobile antibiotic resistance genes” with great interest. The

authors have re-evaluated the evidence for claims of the origin of mobile ARGs by appealing to both the literature and examination of available genome synteny and by synthesizing the approaches into a set of criteria. I believe this is an important and timely topic of interest to many readers in antimicrobial resistance and has the potential to be cited by others in this area.

As the manuscript stands, I cannot recommend publication without some additional rigor in the methods and materials section (see specific comments) that would enhance reproducibility and allow interested parties to follow along more closely. In particular, phylogenetic trees were produced, but it is not clear if these are trees of genomic contexts (nucleotide data), or ARG sequences (protein sequences), or both, but there is no mention of these trees in the results. Similarly, virtually none of the genomic synteny results are presented except in summary, making these results difficult to evaluate - can the genomic synteny results be provided as supplementary data? I have no sense for how many loci or different species are represented for each ARG.

The discussion firmly situates this work in the larger context of the (recent) origins of ARGs. Given the that the ARGs with putatively identifiable origins are restricted to Gram-negative organisms in general, and Proteobacteria specifically, I believe it would be useful for the authors to comment on whether these criteria may be modified to accommodate the 96% of mobile ARGs for which origin cannot currently be assigned.

Specific comments

8. Lines 27-33: I would like some additional citations here - I am not familiar with ISCR and it would be helpful to be pointed to some reviews of this topic to orient the reader.

We agree on the usefulness of additional references on the topic of IS and ISCRs, and have provided them in line 31-32.

9. Line 43: Dan Andersson, in particular, has contributed much to this topic in the literature. Of particular relevance for this you may cite Andersson and Hughes Nat Rev Microbiol 2010;8(4):260-271.

The article is indeed of relevance, so we have referenced it in line 43.

10. Lines 73-79: How many articles were identified by this literature search procedure? How many matching articles were retained? The reference list in the supplementary file 1 contains 77 citations. Having performed a similar search in the past, I have noticed that many articles using the phrase 'origin' report no such thing.

The search using the keywords specified in line 73-74 produces 3342 results. As correctly stated by the reviewer, the great majority of these hits does not report genus or species origins. We went through all hits and filtered articles of potential interest (we now describe these in the manuscript, line 74), which we studied further. In several instances, we found references in such articles which led us to other articles reporting genus/species origins. Based on this, we identified 42 articles suggesting a specific genus/species as origin of a specific (group of) ARG(s). Note that the goal here was not to scrutinize single articles, but rather to scrutinize the collective evidence on the reported ARG origins - So several articles providing evidence for a single ARG origin may be discussed together. Indeed, 77 (79 in the revised version) references are cited in supplementary file 1. However, these do not only contain the articles reporting origins, but also articles reporting expression of the respective ARGs in both their origin and mobile genetic elements (if that information was available), samples from which different genera/species have been identified, ect., as these details are also discussed in supplementary file 1.

In our opinion, the difficulty of identifying these articles highlights the need for a collected resource of such articles, which we, among our other results, aim to provide here.

We also added a short abstract to the results section (line 154-156) describing the number of articles identified and retained ("Our literature search yielded 3342 articles. After screening of all hits (including the references of articles deemed relevant), 43 articles providing evidence for genus or species origins of an ARG were retained for further analysis.").

11. Lines 81-86: How often did an article claim to source the origin of an ARG without being associated with a MGE?

We are not 100% sure that we understood the question. If the reviewer refers to how often articles assigned origin species in which there were no associations of the ARG with MGEs, most had generally no such association (n=31), but a few (n=5) proposed origins did. Most of these ARGs (n=4) were among those where we found there was not sufficient evidence to define a recent origin.

12. Line 92: is the pipeline available for review?

We have added a code availability statement to the revised manuscript in line 461-464. The pipeline and usage instructions are available for review at https://github.com/EbmeyerSt/ARG_loci_comparative_pipeline (We recommend following the tutorial provided at https://github.com/EbmeyerSt/ARG_loci_comparative_pipeline/wiki/Tutorial). While we

of course want to be completely transparent with how we performed our analyses, we want to emphasize that we do not attempt to provide a user-friendly, production level software in this manuscript. The time it would take to make the code ‘production ready’ would be >8 months, and the additional descriptions required would exceed the scope of this manuscript.

13. Line 95-96: How were these novel ARGs identified? In the supplementary file?

We have changed ‘novel’ to ‘recently characterized’ in the respective line, to avoid misunderstandings. The recently characterized ARGs LMB-1, FosA8, FosL (which is not of relevance for this paper, but a part of the database) and GPC-1 were originally identified from plasmids of clinical isolates resistant to the respective antibiotic class. We obtained accession numbers from the original articles (referenced in supplementary file 1).

14. Line 95: What components of CARD were downloaded? I assume the protein sequences of all antibiotic resistance determinants? I believe these are also versioned, so which version was used? Similarly, the ResFinder database is also versioned through the bitbucket repository

(https://bitbucket.org/genomicepidemiology/resfinder_db.git), which version was used?

We used all protein sequences contained in CARDs protein homolog model, in order to avoid including resistance mutations, transcription factor sequences ect, which also are contained in CARD. We have added this information and the respective version CARD at line 94-97 in the manuscript. For ResFinder, we did not use the provided ResFinder pipeline, but only the database files, which are not clearly versioned to the best of our knowledge. Therefore we have added the date we downloaded the database. We provide that information now at line 93-97 of the manuscript (“[...]sequences from CARDs¹⁸ protein homolog model (v3.0.5, downloaded September 2019, recently characterized ARGs LMB-1, FosA8, FosL and GPC-1 were manually added) was mapped against the Resfinder database¹⁹ (downloaded September 2019) using DIAMOND at a 99% identity threshold and a 90% query coverage cutoff”).

15. Line 97: the ‘resulting database’ is the intersection of CARD and ResFinder db by alignment, or the union, to include everything in at least one of the databases? I want to understand what the mapping/alignment procedure between the two databases accomplished.

The resulting database is the intersection of CARDs protein homolog model and ResFinder by alignment using DIAMOND at a 99% identity threshold over 90% of the

sequence length. This was done in order to create a database containing only mobile ARGs (as ResFinder aims to provide) with a clear and structured naming of each sequence (as present in CARD).

16. Line 98: to clarify, 80% query coverage cutoff?

Yes, 80% query coverage cutoff. We have now clarified this in line 97–99 in the revised manuscript (“. Using the resulting database, all genomes and plasmids were searched for the ARGs using DIAMOND blastx v0.9.24.125 with a 70% identity cutoff and an 80% query coverage cutoff”).

17. Line 99–100: these cutoff criteria are relaxed with respect to the database vs database cutoff criterion, were you only including hits in closely related genera, or more distant taxa? It’s very curious to me that the resulting set were restricted entirely to Proteobacteria.

The criteria for annotating ARGs are relaxed on purpose. A 70% identity and 80% query coverage cutoff allows us to investigate whether genes that are more distantly related ($\geq 70\%$ amino acid identity) to a certain ARG are present in taxa closely related to the suspected origin, thus supporting a long-standing association of the respective ARG with the respective genus’ chromosome (e.g the *Aeromonas* AmpC^{3,4}, qnrB in *Citrobacter*⁵). All hits with a sequence similarity $\geq 70\%$ amino acid identity to the respective ARG were included, irrespective of taxon. We have now clarified this in line 99–100 (“This relaxed cutoff was used in order to also identify potentially related genes in other taxa”). We were intrigued as well to find that all origin species were Proteobacteria, 95% previously associated with infection of humans and/or domestic animals.

18. Line 100–102: Genbank assemblies vary greatly with respect to contiguity and completeness - did you have any criteria for excluding highly fragmented assemblies? 10kbp upstream and downstream might not be available in such assemblies, particularly if your goal is to identify ARG associations with IS and Tn sequences which can be highly repetitive and difficult to assemble (and subject to a higher chance of misassembly).

We thank the reviewer for pointing out that we missed to describe this criterion.

As rightly pointed out, short sequences may lack to provide information on mobility/non-mobility. We therefore excluded all contigs (on which ARGs had been identified) with less than 6 ORFs. We now describe this in line 105–107 of the revised manuscript (“. To reduce the number of uninformative sequences (with regards to mobility/non-mobility) for further analysis, sequences containing less than 6 predicted ORFs were excluded”).

19. *Line 120: FastTree or FastTree2? Which version, and what parameters were used to estimate these trees (substitution model, CAT approximation and categories?) Why produce trees only to never refer to them in the remainder of the manuscript? Were the datasets so large that non-approximate ML were infeasible?*

20. **From general comments:** *In particular, phylogenetic trees were produced, but it is not clear if these are trees of genomic contexts (nucleotide data), or ARG sequences (protein sequences), or both, but there is no mention of these trees in the results. Similarly, virtually none of the genomic synteny results are presented except in summary, making these results difficult to evaluate - can the genomic synteny results be provided as supplementary data? I have no sense for how many loci or different species are represented for each ARG.*

We apologize for not describing the used parameters previously. FastTree v2.1.9 was used using a generalized time reversible model with default CAT approximation (20). We have now added the used parameters to the revised manuscript at line 127-128 (“[...]a phylogenetic tree was created using FastTree v2.1.9²⁴ (gtr model, default CAT approximation)”). Some genes (e.g CTX-M, SHV) are found on >2000 unique loci - making non-approximate maximum likelihood methods unfeasible due to the computational time needed. The phylogenies were created based on the alignment of the complete sequences surrounding the ARG (up to 20kbp), and are part of the code used for visualizing the genetic contexts of each sequence containing an ARG - where the tree serves as anchor for placing sequences in a certain order in the final visualization (as briefly described in line 129-131 in the revised manuscript, see answer to next comment). Therefore, these trees by themselves are hard to interpret (especially if large chunks of the origin chromosome have been mobilized, very short sequences are present, or the origin species also contains a plasmid with the respective mobile ARG), and do not contribute evidence that is not also provided by the comparative analysis (both synteny and nucleotide identity). To avoid potentially confusing the reader, we decided to not include those phylogenies in the manuscript. They can however be reproduced by the provided code.

The synteny figures for each gene are just a first step in the comparative genomic analysis. Though they aid in visual comparison of ARG loci from different genomes, helping to decide which genomes to analyze more closely, they are not perfect - taxonomic misclassifications are common in Genbank, and assembly errors may make ORFs unrecognizable for e.g tools predicting protein coding genes. Thus, while useful for orientation, careful manual revision and analysis of the sequences underlying these synteny figures (using e.g manual blast, ISFinder etc) is needed in order to arrive at a conclusion about an origin, as described in the material and method section of the manuscript. The sheer number of ARG loci for certain genes (e.g CTX-M is present in >25000 genomes/plasmids and in >2600 unique loci)

makes these figures unsuitable to orient a reader who is unfamiliar with the specific ARG-locus. We now provide access to the synteny figures at https://figshare.com/articles/figure/ARG_syntenies/13084553 (together with a short README.txt file, providing additional explanations), which also are reproducible by the provided code (given sufficient computational resources).

*21. Line 148-149: I appreciate the distinction made between evolutionary time and functional time. I fear some readers might be confused between the origin of a gene, or the origin of a gene as a mobilized ARG which I believe is the intention in this manuscript. There is some debate in the literature between the idea of a ‘proto-resistance’ gene (Morar and Wright. *Annu Rev Genet* 2010;44:25-51) and a resistance gene being strictly a gene that confers resistance to a clinically relevant antibiotic in the context of a pathogen (Martínez et al, *Nat Rev Microbiol* 2015;13:116-123).*

We are happy about the reviewers’ appreciation. In response to this comment, we have clarified our definition of ‘recent origin’ in line 158-162 of the revised manuscript (“[...]we use the term **recent origin**, describing the bacterium belonging to the evolutionary most recent taxon where the ARG is widespread, but commonly not associated with (any of) the mobile genetic element(s) (e.g. IS, integron, transposon, plasmid or as part of an integron) that played a role in the ARGs transition to its clinically relevant contexts.”).

22. Line 157: I find this idea intriguing, that a potential ARG may be mobilized within a restricted set of taxa

We also find this interesting and think more research on what makes a potential ARG ‘compatible’ with different species is needed.

*23. Line 177-179: In your view, are these criteria, particularly #5, likely to hold true outside of Proteobacteria, and what constitutes independence in related taxa? See Pawlowski et al *Nat Comm* 2016;7:13803 for an investigation of the genetic context/synteny of ARGs in *Paenibacillus*.*

If the mobilization of the ARG was an evolutionary recent event, we are positive that our criteria would also be applicable outside of Proteobacteria. Pawlowski et al 2016 show that the genetic context of CpaA-like enzymes (figure 5c, appended below) is to some extent conserved between the different *Paenibacillus* isolates, whereas the genetic context of VatI-like enzymes is only conserved between isolates that are more closely related to one another (The exception being *P. lactis* and *P. sp. LC231*, which is puzzling given the degree of ARG conservation -

what do other genomes very closely related to these ones look like?). However, when looking at the conservation of both enzymes, figure (5a) shows that CpaA-like enzymes overall appear more conserved between different species compared to VatI-like enzymes - Thus it is not surprising that the genetic contexts of CpaA-like enzymes are more conserved between species than the contexts of VatI-like enzymes. If we now imagine CpaA from *Paenibacillus lactis* being mobilized by e.g ISCR1, our criteria would hold true, also number five. If VatI was to be mobilized from *P. lactis* by ISCR1, our pipeline would not even detect the VatI-like enzymes in most *Paenibacillus* species (except *P. lactis* and *P. sp. LC231*) referenced in figure 5, because they are too dissimilar in amino acid identity from the reference *P. lactis* VatI (<70%), indicating that these enzymes have diverged from the newly mobilized gene in more ancient times. Overall, figure 5 shows that more closely related species have similar ARG-loci (variation between species to some extent is expected, another good example would be the *Aeromonas* AmpC-locus^{3,4}). Thus, if mobilization of the ARG is recent, our criteria would also hold true outside Proteobacteria. We discuss the case of non-recent mobilization and HGT in line 392-399 of the revised manuscript.

As rightly questioned by the reviewer, ‘independent species/strains’ is confusing in this formulation, so we removed it.

Figure 5: The conservation of resistance over millions of years.

24. Line 202: Unsurprising, Proteobacteria are not known as producers of the antibiotics that are or derived from natural products. Fluoroquinolones are synthetic, however, so we would never expect any of these to be producers and this is worth pointing out.

We agree with the reviewers comment and have added this reflection in line 226–228 of the revised manuscript (“None of the here reported recent origins are known antibiotic producers (though this was expected for e.g. the origins of *qnr* genes, as fluoroquinolones are synthetic antibiotics not produced in nature)”).

25. Line 208–209: Given the historically biased nature of the sequence databases towards organisms of medical interest, this isn’ t surprising.

Indeed, when looking at the total number of genomes in Genbank, the majority is derived from pathogens. When looking at the number of unique species however, the situation is different. As described in the results of our overrepresentation analysis (line 240-245), of 100 randomly sampled proteobacterial species only 24 had been reported in infections of humans or animals in the literature, whereas 76 had not. However, out of 22 species origins, 21 have been reported in infection. As described in line 240-245, species that can cause infection are thus much more frequently the origin of an ARG than would be expected by random chance!

26. Line 250-251: I do not know if the authors can say these are correct determinations, only that their criteria support the initial determination of origin.

We agree to the reviewers comment and have modified the respective sentence and replaced ‘correct determinations’ with ‘strong evidence’ .

27. Line 340: Can you provide an example, citation of this? Most of the examples in Table 1 have origins in a particular species, but several only have origins in a genus. Are the factors preventing inference of origins related to the level of sequence diversity of the ARG, the level of genomic diversity in host species, the degree to which an ARG has proliferated outside of its origin taxa, or something else?

We think that finding the origin of resistance genes is often a question of genome availability and hypothesize that the more species are characterized and sequenced, the more origins (and more precise, e.g species rather than genus) we will find. The lack of available well classified genomes is in our experience the greatest factor preventing inference of origins - For several species that only have an origin genus assigned the ARG-variants found in the chromosomes of potential origins are either too divergent from the mobile ARG in sequence identity to confidently assign an origin (e.g GPC-1/BKC-1 from *Shinella* spp. - While GPC-1/BKC-1-like enzymes are present in all 16 to date available *Shinella* spp genomes, synteny at the locus is conserved ect., the closest variant in *Shinella* is 93% identical to mobile GPC-1. From what we observe in other ARGs where we are confident about the origin, this is enough to assign the origin genus, but not the species!), or the species classification within the genus is insufficient because of its novelty (e.g PER variants from *Pararheinheimera* spp. - at the time of publishing, only 3 PER-positive *Pararheinheimera* genomes with unconfirmed species names were available (max 96% nucleotide identity to PER). Now, nearly two years later, more genomes that further confirm this origin have been made available). If an ARG has a recent origin and the origins genome is available, it should be possible to identify it with help of the here formulated

criteria independent of the ARGs sequence diversity or how far the ARG has proliferated outside its origin taxon (see e.g. CTX-M⁶).

28. Line 369: *all to data* -> *all to date*

We thank the reviewer for pointing out this typing error and have now corrected it.

Further changes

We noticed that we had previously missed to describe a clustering step in our visualization pipeline that is implemented in order to reduce the amount of displayed sequences. We now describe this step in line 125-127 (“To reduce the number of sequences to visualize, duplicates were removed. The remaining extracted sequences were then clustered at a 95% nucleotide identity threshold using USEARCH v8.0.1445²². The resulting centroids[...]”).

We have adjusted Figure 1 accordingly (‘bu adding USEARCH to the figure’), and furthermore replaced ‘RAXML/Figtree’ with ‘FastTree’ , as ‘RAXML/FigTree’ was based on a previous version of the used pipeline.

1.) ARG identification

- CARD
- ResFinder

Identification of ARG in genomes and plasmids

DIAMOND

2.) Context extraction & annotation

- Python
- PROKKA
- UniProtKB
- ISFinder
- NR
- IntegronFinder

Extraction of up to 10kbp upstream and downstream of ARG, ORF prediction, gene annotation, integron identification

3.) Comparative analysis

- MAFFT
- USEARCH
- FastTree2
- blastn

Alignment of extracted ARG loci, phylogeny and detailed comparison of nucleotide identities, synteny and host taxa

Criteria evaluation

Fig.1: Comparative genomics workflow and tools/databases used for amendment and scrutiny of proposed origins of ARGs.

We have changed the database used for the overrepresentation analysis from NCBI taxonomy to Genbank assembly in order to be conservative and include only species with publicly available genomes in the analysis. This changes the contents of supplementary file 2, figure 4 and the results of Fishers exact test in line 239 (“(Fishers’ exact test odds ratio: 54.0, $p=3.08e-9$).”) and line 245 (“Fishers’ exact test odds ratio: 66.5, $p=3.13e-10$ ”). These changes do not change any of the conclusions.

Fig. 4: Grouped barchart showing number of species reported in infection for origin and non-origin species (n=122). Non-origin species were randomly selected from all proteobacterial species in the Genbank Assembly database. Fishers’ exact test odds ratio: 66.5, $p=3.13e-10$.

We have furthermore identified an article suggesting an origin for the mobile *fosA1* (previously called *fosA*) gene in *Enterobacter cloacae*. We have added the article to our analysis (shown in table 1 and supplementary file 1).

We have also added a data availability statement (456-459) and a code availability (line 460-464) statement at the end of the manuscript.

References

1. Enault F, Briet A, Bouteille L, Roux S, Sullivan MB, Petit MA. Phages rarely encode antibiotic resistance genes: A cautionary tale for virome analyses. *ISME J* 2017; **11**: 237 - 47. Available at: <http://www.ncbi.nlm.nih.gov/genomes/Geno>. Accessed September 21, 2020.
2. Ghosh S, Sadowsky MJ, Roberts MC, Gralnick JA, LaPara TM. *Sphingobacterium* sp. strain PM2-P1-29 harbours a functional *tet* (X) gene encoding for the degradation of tetracycline. *J Appl Microbiol* 2009; **106**: 1336 - 42. Available at: <http://doi.wiley.com/10.1111/j.1365-2672.2008.04101.x>. Accessed September 8, 2020.
3. EBMEYER S, KRISTIANSOON E, LARSSON DGJ. The mobile FOX AmpC beta-lactamases originated in *Aeromonas allosaccharophila*. *Int J Antimicrob Agents* 2019. Available at: <https://www.sciencedirect.com/science/article/pii/S0924857919302687?via%3Dihub>. Accessed October 14, 2019.
4. Ebmeyer S, Kristiansson E, Larsson DGJ. CMY-1/MOX-family AmpC β -lactamases MOX-1, MOX-2 and MOX-9 were mobilized independently from three *Aeromonas* species. *J Antimicrob Chemother* 2019. Available at: <https://academic.oup.com/jac/advance-article/doi/10.1093/jac/dkz025/5309034>. Accessed February 13, 2019.
5. Ribeiro TG, Novais Â, Branquinho R, Machado E, Peixe L. Phylogeny and Comparative Genomics Unveil Independent Diversification Trajectories of *qnrB* and Genetic Platforms within Particular *Citrobacter* Species. *Antimicrob Agents Chemother* 2015; **59**: 5951 - 8. Available at: <http://www.ncbi.nlm.nih.gov/pubmed/26169406>. Accessed January 30, 2018.
6. Cantón R, González-Alba JM, Galán JC. CTX-M Enzymes: Origin and Diffusion. *Front Microbiol* 2012; **3**: 110. Available at: <http://www.ncbi.nlm.nih.gov/pubmed/22485109>. Accessed January 30, 2018.

REVIEWERS' COMMENTS:

Reviewer #2 (Remarks to the Author):

I found all of my comments to be thoroughly addressed and think that the revised manuscript is significantly improved in its clarity. I do think it might be worthwhile for the authors to read and possibly incorporate/discuss (potentially in the context of line ~373) Jiang et al 2017 Nat Comm "Dissemination of antibiotic resistance genes from antibiotic producers to pathogens" (though it is possible the authors have already done this and I missed it).

Aside from this I only noted a few typographical errors and think the manuscript is in very good shape.

Line 108 less -> fewer

Line 222 tetX -> TetX

Line 293 need an 'is' in there

Line 302 has -> have

Line 336 an ARG -> a mobilizable ARG

Line 370 many novel -> many novel

Line 399 Delete one instance of 'often'

Reviewer #3 (Remarks to the Author):

Having read the revised manuscript from Ebmeyer et al I would like to express appreciation for the effort the authors put into their revisions. The additional details, particularly to the methods and provision of the pipeline will clearly help readers follow along/reproduce this work.

Specific comments related to numbered points in the rebuttal:

10-11. I think these answers, in combination with the response to point 4 for referee #2, adequately clears up some confusion about the work the word 'origin' is performing – to recognize an ARG's origin as a mobilized element vs where (and when) an ARG first arises. Literature claiming 'origins' sometimes rely on preconceived notions of the direction in which genes move. If the perception is that a pathogen species is the destination of an ARG, sometimes the non-pathogen species is automatically considered the origin even in cases where the gene is not associated with a mobile element.

12. I appreciate that no claim was made for the presentation of production quality software, but in as far as the logic of the methods are represented by steps in software I find it sometimes easier to follow along, so much thanks.

17. As referee #2 raised the same issue in point 3, it seems clear to me this is a bias towards the increased diversity and attention paid to Proteobacteria in the sequence databases.

20. This was a lot of work, and I appreciate this greatly, thank you.

Reviewer #2 (Remarks to the Author):

1. I found all of my comments to be thoroughly addressed and think that the revised manuscript is significantly improved in its clarity. I do think it might be worthwhile for the authors to read and possibly incorporate/discuss (potentially in the context of line ~373) Jiang et al 2017 Nat Comm "Dissemination of antibiotic resistance genes from antibiotic producers to pathogens" (though it is possible the authors have already done this and I missed it).

We are glad to see that the reviewer is happy with our revisions. In response to this comment, we have incorporated a short discussion on the results of the mentioned study in line 227-232.

2. Aside from this I only noted a few typographical errors and think the manuscript is in very good shape.

Line 108 less -> fewer

Line 222 tetX -> TetX

Line 293 need an 'is' in there

Line 302 has -> have

Line 336 an ARG -> a mobilizable ARG

Line 370 many novel -> many novel

Line 399 Delete one instance of 'often'

We are thankful the reviewer pointed out these errors and have corrected them.

Reviewer #3 (Remarks to the Author):

Having read the revised manuscript from Ebmeyer et al I would like to express appreciation for the effort the authors put into their revisions. The additional details, particularly to the methods and provision of the pipeline will clearly help readers follow along/reproduce this work.

Specific comments related to numbered points in the rebuttal:

3.

10-11. I think these answers, in combination with the response to point 4 for referee #2, adequately clears up some confusion about the work the word 'origin' is performing – to recognize an ARG's origin as a mobilized element vs where (and when) an ARG first arises. Literature claiming 'origins' sometimes rely on preconceived notions of the direction in which genes move. If the perception is that a pathogen species is the destination of an ARG, sometimes the non-pathogen species is automatically considered the origin even in cases where the gene is not associated with a mobile element.

We are happy that the reviewer thinks that the clarity of our definition is adequately cleared up. We thank once again for this important comment.

4.

12. I appreciate that no claim was made for the presentation of production quality software, but in as far as the logic of the methods are represented by steps in software I find it sometimes easier to follow along, so much thanks.

We are glad the reviewer is content with our provided code.

5.

17. As referee #2 raised the same issue in point 3, it seems clear to me this is a bias towards the increased diversity and attention paid to Proteobacteria in the sequence databases.

Indeed, and we address the bias towards proteobacterial species in Genbank in line 224-227 in the revised manuscript.

6.

20. This was a lot of work, and I appreciate this greatly, thank you.

We are happy the reviewer appreciates our effort and are in turn grateful for the reviewers' constructive comments on the manuscript.